# Direct observation of organic molecules in asteroid ryugu revealed by high-resolution atomic force microscope

Kota Iwata [1] ✉, Yasuhiro Oba [2], Hiroshi Naraoka [3], Hikaru Yabuta [4,5], Shogo Tachibana [6,7] & Yoshiaki Sugimoto [1,5] ✉

A diverse variety of organic matter exists in space. Since these extraterrestrial organic materials preserve chemical information from the early solar system, their identification has been extensively studied. However, detailed structural information has remained limited. Here we investigate organic matter from the carbonaceous asteroid Ryugu using a high-resolution atomic force microscope (AFM). We directly resolve the chemical structures of individual organic molecules from the asteroid. We find a wide variety of polycyclic aromatic hydrocarbons (PAHs), many of which are unexpectedly large in size. The largest one is composed of approximately 100 fused rings, significantly larger than the extraterrestrial PAHs identified in previous ensemble-level analyses. These PAHs exhibit non-planar structures incorporating five-, six-, seven-, and even eight-membered rings. Such complex structures can be resolved in detail only through single-molecule AFM analysis.

Organic matter in outer space preserves ancient information about processes such as molecular cloud evolution during star formation and the early stages of the birth of the solar system[1]. It has also been proposed that the delivery of such organic materials to Earth contributed to its habitability and even the origin of life[2]. To explore the nature and formation pathways of extraterrestrial organic matter, researchers have mainly studied organic compounds extracted from carbonaceous meteorites[3]. A typical carbonaceous meteorite is composed mostly of mineral components but contains about a few percent of organic matter relative to its mass. The organic matter is chemically extracted and separated into insoluble organic matter (IOM), which is insoluble in solvents and accounts for more than 70 wt% of the total organic matter, and soluble organic matter (SOM), which is soluble in various solvents. The extracted organic matter is evaluated by various analysis methods.

For example, the SOM of the Murchison meteorite, one of the most well-studied meteorites, contains approximately 50,000 chemical compositions identified by high-resolution mass spectrometry[4]. A wide range of organic structures in the SOM have been identified, including amino acids[5,6], carboxylic acids[7,8], aliphatic chains[9], nucleobases[10,11], polycyclic aromatic hydrocarbons (PAHs)[12,13], and sugars[14].

In the case of conventional methods that rely on meteorites that have fallen to Earth to obtain extraterrestrial organic matter, there is a possibility of contamination by the Earth's atmosphere, soil, or organisms before they are recovered. Recent sample return missions have successfully brought back samples from the C-type asteroid Ryugu in 2020 and from the B-type asteroid Bennu in 2023. Unlike meteorites that fell to Earth, these fresh extraterrestrial samples avoided terrestrial exposure and were handled under strict curation protocols[15], making them essentially contamination-free.

By analysing all SOMs of Ryugu and Bennu by Fourier transform-ion cyclotron resonance mass spectrometry, approximately 23,000 chemical compositions in Ryugu[16] and 13,000 in Bennu[17] have been

[1]Department of Advanced Materials Science, University of Tokyo, Kashiwa, Japan. [2]Institute of Low Temperature Science, Hokkaido University, Sapporo, Japan. [3]Department of Earth and Planetary Sciences, Kyushu University, Fukuoka, Japan. [4]Department of Earth and Planetary System Science, Hiroshima University, Higashi-Hiroshima, Japan. [5]International Institute for Sustainability with Knotted Chiral Meta Matter (WPI-SKCM2), Hiroshima University, Higashi-Hiroshima, Japan. [6]Department of Earth and Planetary Science, University of Tokyo, Tokyo, Japan. [7]Institute of Space and Astronautical Science, Japan Aerospace Exploration Agency (JAXA), Sagamihara, Japan. ✉e-mail: kiwata@g.ecc.u-tokyo.ac.jp; ysugimoto@k.u-tokyo.ac.jp

detected. Among them, organic molecules such as amino acids, carboxylic acids, nucleobases, PAHs, and nitrogen-containing cyclic compounds have been detected[18,19]. The amino acids were detected in racemic form, suggesting their extraterrestrial origin. It has been reported that phenanthrene (three-rings) and pyrene (four-rings) are dominant among the PAHs in Ryugu samples (A0106 and C0107)[18]. Furthermore, isotope analysis of PAHs in the Ryugu sample suggested that one of the possible formation processes is their production in interstellar molecular clouds[20]. Recently, two-step laser mass spectrometry (L2MS) analysed the Ryugu sample in its powdered form[21]. Although this analysis showed characteristics similar to those of SOM, it also implied the presence of trace amounts of large aromatic molecules composed of more than 60 carbon atoms. This study further suggested that the series of alkylated species, ranging from methyl-substituted to longer chains, appears to be present in the Ryugu sample. To date, numerous chemical species have been detected in extraterrestrial samples as described above. However, structurally identified molecules have been limited to relatively abundant and small species, whereas molecules with low abundances or large and complex structures remain technically challenging to detect and identify.

In contrast to laboratory analyses of extraterrestrial samples, infrared spectroscopic observation has indicated that large PAHs are present in interstellar regions. Earlier studies suggested that PAH consisting of more than 100 carbon atoms may exist and proposed the possible structures for PAHs containing more than 50 fused rings[22]. However, more recent analyses incorporating anharmonicities in PAH infrared spectra suggested that typical PAH sizes may instead fall within the range of 40–50 carbon atoms[23]. Nevertheless, theoretical models of interstellar PAH population predicted that PAHs extending to approximately 100 carbon atoms may be present in the interstellar medium[24]. Thus, there has been a gap between PAHs observed in the interstellar regions and those identified in planetesimal samples.

The frequency modulation atomic force microscope (FM-AFM) is a family of scanning probe microscopes that can image surface topography at the single atomic/molecular scale by measuring the force between the tip and the sample. In 2009, it was reported that the chemical structure of organic molecules can be directly imaged by using a CO-functionalized tip and measuring the repulsive force between the tip and molecules[25]. Today, this technique is routinely used to resolve the chemical structures of various organic molecules. It has enabled the structural identification of natural products with unknown structures, such as bacterial metabolites[26], asphaltenes[27,28], organics dissolved in seawater[29], and soot[30,31]. This ability to resolve the

chemical structure of molecules can be applied as a new technique to bring complementary information to traditional measurement methods of astrochemistry.

High-resolution AFM imaging has recently been applied to the identification of astrochemical samples. For example, an analog sample synthesized under an environment simulating Saturn's satellite Titan was measured, and the chemical structure of PAHs containing aliphatic chains and nitrogen atoms was directly visualized[32]. In addition, AFM observation of the products from the gas-phase growth of carbon and hydrogen under conditions mimicking a circumstellar envelope verified the synthesis pathway of alkanes found in meteorites[33]. Furthermore, AFM measurements of the SOM of the Murchison meteorite as a natural extraterrestrial sample directly confirmed the presence of organic molecules such as pyrene ($C_{16}H_{10}$) and aliphatic chains that are consistent with previous mass spectrometry and other results[34]. Although AFM is a promising tool to identify the structure of astrochemical organic samples, its application has so far been limited to confirming expected molecular structures.

In this study, we investigated Ryugu organics extracted by dichloromethane (DCM), adsorbed on a metal substrate, and characterized them by high-resolution AFM observation. As a result, we identified a wide variety of PAHs. All observed molecules had distinct structures, ranging from PAHs with about six fused rings to giant PAHs estimated to contain more than 100 fused rings. These structures are markedly different from those of the SOM detected in the previous mass spectrometry analyses that identified PAHs composed of one to six fused rings[18]. By using single-molecule-level AFM analysis, we were able to identify the existence of such molecules.

## Results and discussion
### Real space structural identification

We investigated the DCM extract of the SOM from the Ryugu sample, because initial analysis showed that it is PAH-rich[18]. Figure 1 shows an example of structural identification of a molecule from the Ryugu sample by AFM. Figure 1a shows the scanning tunneling microscope (STM) overview of the sample prepared by depositing the DCM extract heated to 80 °C (for the details of sample preparation, see "Methods"). The red rectangle indicates a molecule derived from DCM extract, and the depressions are CO molecules. Typically, we observed only a few Ryugu-derived molecules in a 2 μm × 2 μm area. Figure 1b enlarges the molecule indicated by the red rectangle in Fig. 1a, imaged with a CO tip.

To resolve the chemical structure, we performed AFM constant-height measurement, in which the tip scans parallel to the substrate surface without feedback control of the tip height, as shown in Fig. 1c.

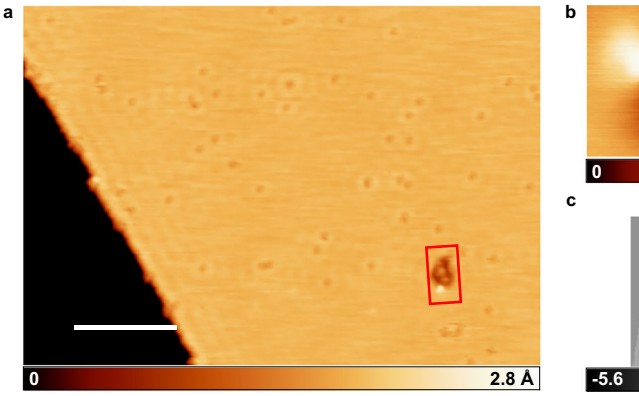
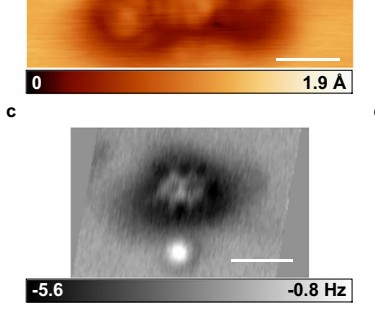
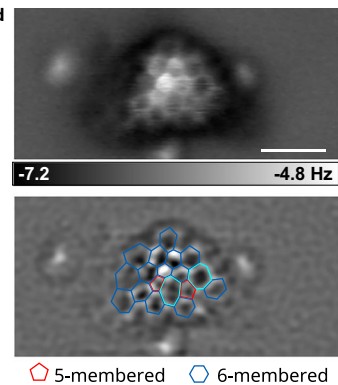

**Fig. 1 | Molecule in the dichloromethane extract of Ryugu sample adsorbed on Cu(111) imaged by scanning tunneling microscope (STM) and atomic force microscope (AFM). a** Overview STM image. The red rectangle indicates a molecule evaporated from the dichloromethane extract of the Ryugu sample. **b** Zoomed in STM image of molecule indicated in (**a**). **c, d** Constant height (**c**) and multi-pass (**d**) AFM image of molecule in (**b**). **e** Image obtained by processing (**d**) to enhance edges. Ring structure is overlaid. Red, blue, and light blue represent five-, six-, and seven-membered rings, respectively. Scale bar in (**a**) is 10 nm, and others are 1 nm.

When the distance between the tip and sample was reduced gradually, a structure consisting of multiple six-membered rings became visible due to repulsive forces. However, comparison with the molecular dimensions obtained by STM indicates that AFM resolved only a part of the structure. This is because only the higher-lying moieties of the molecule are imaged. In the constant-height measurement without feedback control of tip height, lower-lying moieties cannot be resolved (for the detailed description, see Supplementary Information).

Then, we performed the measurement in multi-pass mode, which allows quasi-constant-height measurement even over corrugated structures[35,36]. In this mode, the STM is first scanned with constant-current feedback, and the same line is scanned again along the trajectory of the first scan, with the distance between the tip and samples varied by an arbitrary offset. This approach allows the chemical structure of molecules with three-dimensional features to be observed (see SI). The result is shown in Fig. 1d. Compared with the conventional constant-height measurement (Fig. 1c), a more detailed structure was imaged. This indicates that the molecules lie on the surface in a non-planar structure. Figure 1d was processed by Laplacian filtering to enhance the contrast of the image and is shown in Fig. 1e. The ring structure expected from the AFM image is superimposed. From this image, the molecule is composed of 18 fused rings, including five- (red), six- (blue), and seven- (light blue) membered rings. The presence of heteroatoms in the molecule has been reported for the same Ryugu sample (A0106)[37]. In the previous structural identification of molecules by AFM, heteroatoms such as nitrogen, oxygen, and sulfur have been identified[28,32,34]. However, this identification relies on subtle contrast differences in the AFM images, which is not applicable in our case, where the molecule exhibits a three-dimensional structure that can affect the contrast (for detailed reasons, see "Methods").

Figure 2 shows a detailed structural observation of another molecule from the Ryugu sample. In the STM image (Fig. 2a), a molecule of approximately 3 nm in size is surrounded by multiple bright protrusions. We also performed multi-pass AFM measurement of this molecule. The result is shown in Fig. 2b. As in Fig. 1, the image was processed, and the cyclic structures predicted from the AFM image were superimposed in Fig. 2c. The aromatic core consists primarily of six-membered rings but also includes a small number of five- and seven-membered rings. Notably, even an eight-membered ring (orange) is also present.

The bright protrusion indicated by the arrow corresponds to a CO molecule. Based on their distinct appearance, other bright protrusions around the molecules are not CO molecules. Since the substrate was kept at a low temperature during deposition, it is unlikely that small molecules or impurities aggregated around the molecules due to

thermal diffusion. These bright protrusions are observed only near the aromatic core, suggesting that they represent functional groups attached to the core. A methyl group is a representative example of such a functional group, appearing as a bright protrusion in AFM image[27]. Therefore, the bright protrusions are likely attributable either to methyl groups directly attached to the core or to very short side chains terminated by methyl groups.

## Characteristics of the observed molecules

Figure 3 summarizes all AFM images in which at least part of the chemical structure was observed. In this study, we obtained high-resolution images of 22 molecules, including those shown in Figs. 1, 2. The left panel shows the raw image, and the right panel shows the processed image overlaid with the deduced structural model. The observed molecules ranged from small species with only a few six-membered rings to very large molecules consisting of dozens of rings. No two molecules shared the same structure; all were structurally unique. Multi-pass mode was essential to reveal these details. Consequently, all observed molecules can be distinctly characterized as possessing inherently three-dimensional structures.

The smallest structures consist of about five fused rings (Fig. 3a, u). Figure 3a clearly shows a structure of one six-membered and two five-membered rings, along with two additional unclear rings. In Fig. 3u, the core comprises four six-membered rings and an unclear ring, with many side chains attached. In contrast, the other observed molecules had much larger aromatic cores. Note that this dataset is likely biased toward the moderately larger molecules due to the limitations of our method, as described in the "Methods" section.

The observed cores are mainly composed of six-membered rings. They sometimes contain some five- and seven-membered rings (Fig. 3h, s, etc.). Some molecules had cores in which about half were composed of six-membered rings and the rest of five- and seven-membered rings (Fig. 3g, m). In a few cases, even eight-membered rings were present (Fig. 3k, m, n, p). These non-six-membered rings can be one of the possible sources of three-dimensional structure.

## Estimation of the size of molecules

Here, we estimated the number of rings and molecular weight for each core, under the following assumptions: (i) All ring structures consist of carbon. We ignored possible heteroatoms, such as nitrogen, oxygen, and sulfur, because they cannot be identified reliably as described above. (ii) Regions of the core with unresolved structure were assumed to be six-membered (graphene-like) rings, and we estimated the amount of six-membered rings and carbon atoms from the area of the

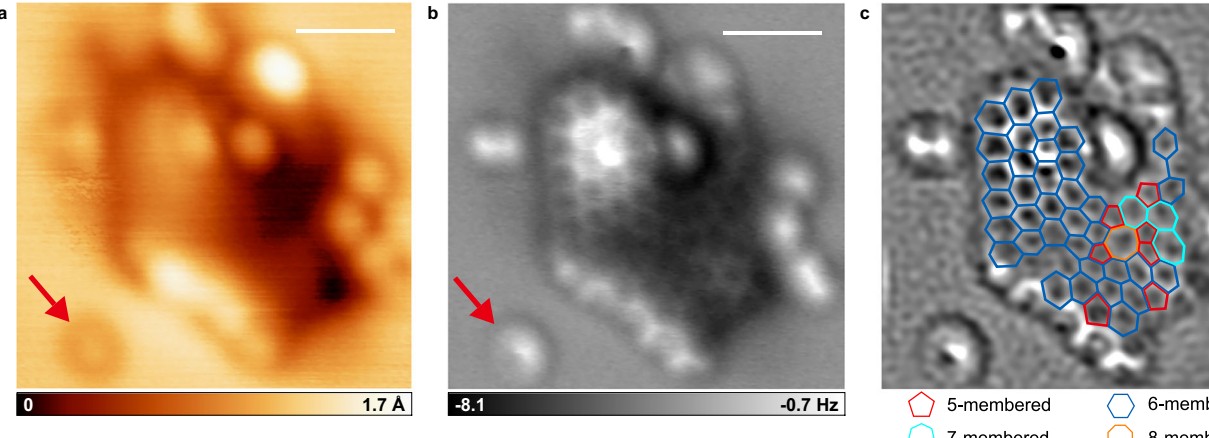

**Fig. 2 | Detailed structural analysis of another molecule in the dichloromethane extract of the Ryugu sample. a** Scanning tunneling microscope image of a molecule from the Ryugu sample. **b** Atomic force microscope image acquired by the multi-pass method. **c** Processed image of (**b**). Ring structure is overlaid. The colors are the same as in Fig. 1, but orange represents an eight-membered ring. Red arrows in (**a**, **b**) indicate a single CO molecule. The scale bars are 1 nm.

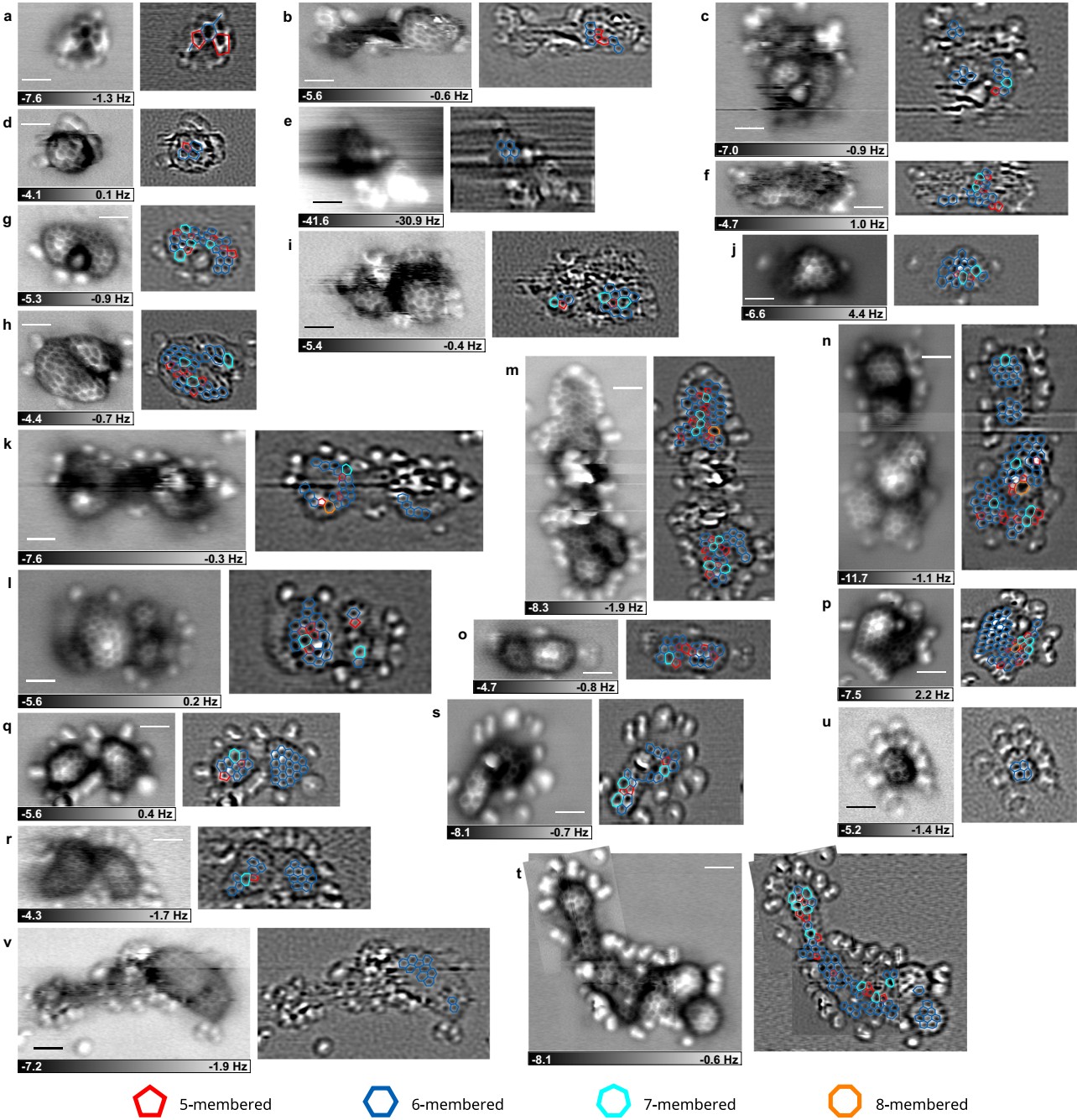

**Fig. 3 | All atomic force microscope (AFM) images of molecules in the dichloromethane extract of the Ryugu sample that successfully resolved their chemical structures. a–v** Left panels show raw AFM images, and right panels show processed AFM images. The structural models are overlaid on the processed AFM images. Red, blue, light blue, and orange represent five-, six-, seven-, and eight-membered rings, respectively. The scale in all images represents 1 nm. **a–l**, **m–v** were prepared by heating the soluble organic matter (SOM) source up to 80 and 110 °C, respectively.

region. (iii) For molecular weight estimation, only carbon rings were considered, and the terminating hydrogens and other elements were ignored. The results are shown in the histogram in Fig. 4. The number of rings ranges from six to over 100, and the molecular weights range from 250 to 3200. Within this dataset, no clear peaks were observed, and there was no indication that molecules of specific molecular weight or ring number were more prevalent.

## Comparison with ensemble-level approaches
Most of the observed PAHs are much larger than extraterrestrial SOM structurally identified by ensemble-level methods[3,16,18]. In an initial analysis of the same Ryugu sample, A0106, a variety of PAHs were found in the SOM by gas chromatography/mass spectrometry (GC/MS)[18,38]. The most abundant PAHs were fluoranthene and pyrene (both $C_{16}H_{10}$), followed by chrysene and/or triphenylene (both $C_{18}H_{12}$), the largest PAHs identified by GC/MS, all of which are composed of four fused rings. Large aromatic cores like those in our study were not identified. One possible reason could be that they were outside the detection range of MS. Generally, SOM is analysed in the range of molecular weight $100 < m/z < 650$[16,18]. In addition, amounts of about $10^{-15} \sim 10^{-18}$ mol are required for detection with current analysers. In contrast, AFM can visualize molecules as long as they adsorb to the

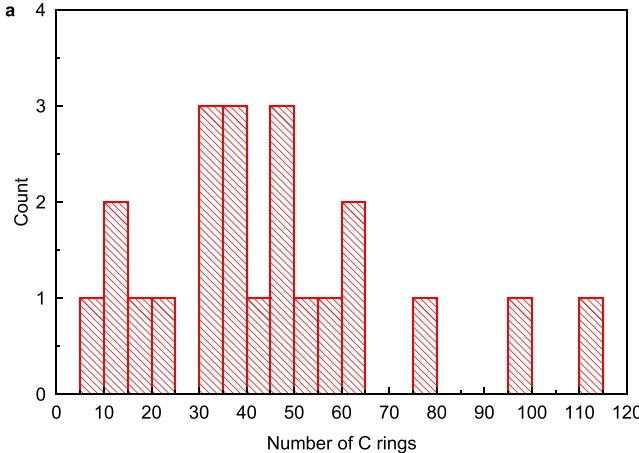

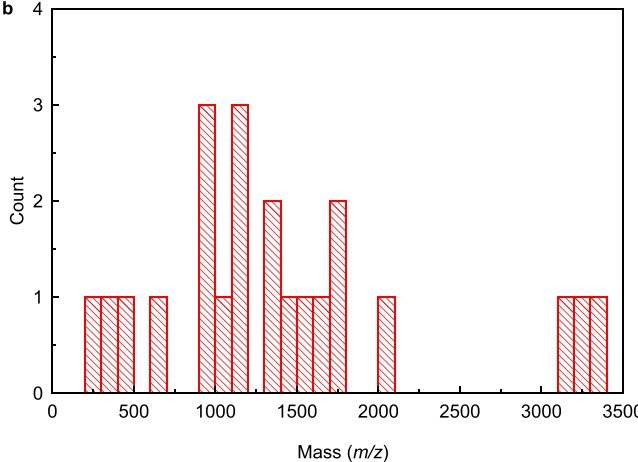

**Fig. 4 | Size distribution of molecules in dichloromethane extract of the Ryugu sample resolved by atomic force microscope. a** Estimated number of six-membered carbon rings and **b** estimated mass of observed molecules shown in Fig. 3.

substrate and the experimenter can find them, regardless of their molecular weight and even if only a single molecule is present.

Recently, L2MS analysis of a different Ryugu sample (C0083) in crushed powder form[21] showed characteristics with a dominant molecular weight of 100–300 and strong peak intensities at $C_{15}H_{11}$ and $C_{16}H_{10}$, which are similar to the initial analysis of SOM. Furthermore, this study also revealed trace amounts of large aromatic molecules, up to $C_{61}H_{44}$ ($m/z = 776$) and $C_{61}H_{47}NO$ ($m/z = 809$). Some of the molecules in our study might share similarities with these molecules, as suggested by the mass spectra. However, even allowing for potential overestimation in our molecular weights analysis, many of the molecules we observed appear larger than these $C_{61}$ species.

Infrared spectroscopies have shown that interstellar PAHs can reach sizes of several tens to one hundred carbon atoms[23,24]. The large molecules observed by AFM may be tentatively associated with such PAHs. Furthermore, all our PAHs have three-dimensional structures. Although the curved molecules like fullerenes have been detected in interstellar and circumstellar regions by infrared spectroscopy[39,40] and in meteorite[41], their formation process of such non-planar molecules remains under debate. Our molecules might relate to intermediates of such species, and their direct identification can shed light on their formation pathways.

Our high-resolution AFM analysis resolved a wide variety of PAHs from the asteroid Ryugu. In particular, we identified unexpectedly large PAHs that have remained undetected, likely because of the detection limits inherent in ensemble-level techniques commonly used in astrochemistry. Many of these molecules exhibited non-planar structures, probably due to the presence of non-six-membered rings and heteroatoms. Such molecules could only be detected through single-molecule-level AFM analysis. Our observation highlights the unique capability of AFM to provide direct structural information at the single-molecule level, complementing the established analytical methods of astrochemistry. We believe that this ability to resolve individual molecules can be extended to other astrochemical samples, such as other organic matter from Ryugu and Bennu samples, and meteorites, and laboratory-synthesized analogs, thereby enabling comparisons among them that can provide important insight into astrochemistry.

## Methods

### Sample preparations of Ryugu DCM extract

The Ryugu sample was provided through the Announcement of Opportunity for Ryugu samples. We analysed the DCM extract of the SOM from the Ryugu sample A0106. The organic matter from the aggregate sample of A0106 (17.15 mg) was extracted sequentially with hexane, DCM, methanol, and water using a sonicator in a Teflon vial, followed by centrifugation. Subsequently, 5 μL of the DCM extract was dropped onto a silicon substrate and dried. It was then mounted on a heating fixture and introduced into an ultra-high vacuum (UHV, ~$10^{-10}$ Torr). Cu(111) was used as the substrate. A clean surface was prepared by several sputtering and annealing cycles, and a small area of bilayer NaCl film was prepared by depositing NaCl, which facilitates CO functionalization of the tip. After confirming the cleanliness of the substrate by STM observation, the substrate below 50 K was brought close to the DCM extract-dropped Si substrate. The DCM extract was deposited onto the sample substrate by heating the Si substrate instantaneously to 80 ~ 110 °C. After the deposition, the substrate was immediately cooled down to 5 K. A small amount of CO for the functionalization of the tip apex was adsorbed on the substrate.

### AFM measurements

An Omicron LT-AFM/STM working in UHV at 4.8 K was used for measurements. The qPlus sensor (typical eigenfrequency, Q factor, and oscillation amplitude were 25.5 kHz, 8000, and 100 pm, respectively) with a W tip was used as an AFM sensor. The tip was prepared by indentation into the Cu substrate and picking up a CO molecule from the substrate for observation of the molecules.

### Limitations in AFM measurement

Our technique has several limitations. First, our deposition method restricts the size of molecules adsorbed on the substrate: small volatile molecules may evaporate in UHV before deposition, and some remaining small molecules may not adsorb on the substrate efficiently due to their low adsorption probability. While we successfully deposited large molecules as shown in the main text, excessively large molecules may not be evaporated at the heating temperature used in this study. As a result, our dataset may indeed be biased toward moderately large molecules. Second, AFM cannot easily resolve the detailed molecular structures. Current elemental analysis of molecules relies on subtle contrast differences in AFM images, which apply primarily to the planar molecules, while three-dimensional structures affect the contrast, rendering this method inapplicable to the three-dimensional molecules, such as those shown in the main text. Third, AFM analysis is intrinsically "slow". Single-molecule level analysis requires locating the target molecule within the μm-scale area, preparing the AFM tip suitable for high-resolution imaging, and performing the imaging. This process takes several hours for the analysis of a single molecule, making it impractical to evaluate a large number of molecules as in ensemble-level analyses.

## Data availability

The SPM data presented in this paper are available from the corresponding authors upon request.

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

## Acknowledgments

We thank A. Kouchi for his support in carrying out this research. We also acknowledge JAXA for the allocation of the Ryugu sample through the 2nd Announcement of Opportunity of Ryugu sample. This work was supported by JSPS KAKENHI (JP20H05849 and JP22H01950 to Y.S., JP23K13665 to K.I., and JP23H00148 to H.N.) and the Asahi Glass Foundation.

## Author contributions

Author contributions are defined according to the CRediT. Conceptualization: Y.S. Methodology: K.I. and Y.S. Resources: H.N., Y.O., and H.Y. Investigation: K.I. Visualization: K.I. Funding acquisition: All authors. Project administration: Y.S. Supervision: Y.S. and S.T. Writing – original draft: K.I. Writing – review & editing: All authors.

## Competing interests

The authors declare no competing interests.
