## [Transparent Peer Review file · Nature Communications]

Direct Observation of Organic Molecules in Asteroid Ryugu Revealed by High-Resolution Atomic Force Microscope

Corresponding Author: Dr Yoshiaki Sugimoto

Version 0:

Reviewer comments:

Reviewer #1

(Remarks to the Author)

I reviewed a previous version of the manuscript and I am very happy to see how well and in detail the authors have answered my queries and addressed my comments.

The improvements made to the manuscript following comments from myself and other reviewers, the manuscript now reads very well and is significantly clearer than the previous iteration. I believe it is worthy of publication in Nat. Comm.

The only comment that I have as that one point I made in my review was not directly addressed. Namely, that of their reference to the review of Tielens from 2008 in support of PAHs containing more than 50 fused rings being present in the ISM based on the AIBs.

There is recent work by Lemmens et al. (DOI: 10.1039/d2fd00180b) where the previous values of $N_C > 50$ are suggested to be overestimated due to the lack of anharmonic calculations in procuring the IR spectra of model PAHs. However, when anharmonicities in the PAH IR spectra based on experimental spectra and state-of-the-art calculations are properly accounted for, this value reduces to $N_C = 40-50$.

Of course, this doesn't exclude larger PAHs being likely present in the ISM and there are older models predicting PAH sizes upward to $N_C = 100$ (see Li & Draine 2001, DOI: 10.1086/323147).

It could be worth including these references to give the reader context of both past and state-of-the-art.

Respectfully yours,
Helgi Rafn Hrodmarrsson

Reviewer #2

(Remarks to the Author)

The study of Kota Iwata et al. used high-resolution AFM to directly image individual organic molecules from asteroid Ryugu, revealing unexpectedly large and structurally complex polycyclic aromatic hydrocarbons that could only be resolved at the single-molecule level.

All points of the previous review were taken seriously and adapted systematically in the new version. The authors also reduced their initial paper to a rather more focused, short and to-the-point message; this is highly appreciated and sharpened overall the manuscript's novelty.

I confirm that the paper is now in a good condition to be accepted for publication.

Response to Reviewers' comments:

We are grateful to all the reviewers for carefully reading our manuscript and confirming the publication in Nature Communications.

Reviewer #1 (Remarks to the Author):

I reviewed a previous version of the manuscript and I am very happy to see how well and in detail the authors have answered my queries and addressed my comments.

The improvements made to the manuscript following comments from myself and other reviewers, the manuscript now reads very well and is significantly clearer than the previous iteration. I believe it is worthy of publication in Nat. Comm.

We sincerely thank the reviewer for the previous comments, which provided us with the opportunity to substantially improve our manuscript. We greatly appreciate your positive evaluation and recommendation for publication in Nature Communications.

Bellow, we answered comments from the reviewer.

The only comment that I have as that one point I made in my review was not directly addressed. Namely, that of their reference to the review of Tielens from 2008 in support of PAHs containing more than 50 fused rings being present in the ISM based on the AIBs.

There is recent work by Lemmens et al. (DOI: 10.1039/d2fd00180b) where the

previous values of $N_C > 50$ are suggested to be overestimated due to the lack of anharmonic calculations in procuring the IR spectra of model PAHs. However, when anharmonicities in the PAH IR spectra based on experimental spectra and state-of-the-art calculations are properly accounted for, this value reduces to $N_C = 40-50$.

Of course, this doesn't exclude larger PAHs being likely present in the ISM and there are older models predicting PAH sizes upward to $N_C = 100$ (see Li & Draine 2001, DOI: 10.1086/323147).

It could be worth including these references to give the reader context of both past and state-of-the-art.

Respectfully yours,
Helgi Rafn Hrodmarsson

We thank the reviewer for pointing out the additional model and more recent work. We have revised both the Introduction and the Discussion sections of the manuscript to incorporate these studies.

In the Introduction section, we have revised the relevant sentences.

We have moved the following sentences to the end of the previous paragraph (P3, L67).

“To date, numerous chemical species have been detected in extraterrestrial samples as described above. However, structurally identified molecules have been limited to relatively abundant and small species, whereas molecules with low abundances or large and complex structures remain technically challenging to detect and identify.”

A new paragraph then begins as follows.

“In contrast to laboratory analyses of extraterrestrial samples, infrared spectroscopic observations have indicated that large PAHs are present in interstellar regions. Earlier studies suggested that PAHs consisting of more than 100 carbon atoms may exist and proposed the possible structures for PAHs containing more than 50 fused rings (Tielens 2008). However, more recent analyses incorporating anharmonicities in PAH infrared spectra suggested that typical PAH sizes may instead fall within the range of 40-50 carbon atoms (Lemmens et al., 2023). Nevertheless, theoretical models of the interstellar PAH population predicted that PAHs extending to approximately 100 carbon atoms may be present in the interstellar medium (Li and Draine, 2001). Thus, there has been a gap between PAHs observed in the interstellar regions and those identified in planetesimal samples.”

In the Discussion section, we have revised the relevant sentences beginning at P8, L205 as follows.

“Infrared spectroscopic studies have shown that interstellar PAHs can reach sizes of several tens to one hundred carbon atoms (Lemmens et al., 2023; Li and Draine, 2001). The large molecules observed by AFM may be tentatively associated with such PAHs.”

Reviewer #2 (Remarks to the Author):

The study of Kota Iwata et al. used high-resolution AFM to directly image individual organic molecules from asteroid Ryugu, revealing unexpectedly large and structurally complex polycyclic aromatic hydrocarbons that could only be resolved at the single-molecule level.

All points of the previous review were taken seriously and adapted systematically in the new version. The authors also reduced their initial paper to a rather more focused, short and to-the-point message; this is highly appreciated and sharpened

overall the manuscript's novelty.

I confirm that the paper is now in a good condition to be accepted for publication.

We sincerely thank the reviewer for the careful evaluation of our manuscript and for the very positive comments. We greatly appreciate your confirmation that the manuscript is now in good condition for publication.